# Malaria Transmission around the Memve’ele Hydroelectric Dam in South Cameroon: A Combined Retrospective and Prospective Study, 2000–2016

**DOI:** 10.3390/ijerph16091618

**Published:** 2019-05-09

**Authors:** Lili R. Mbakop, Parfait H. Awono-Ambene, Stanislas E. Mandeng, Wolfgang E. Ekoko, Betrand N. Fesuh, Christophe Antonio-Nkondjio, Jean-Claude Toto, Philippe Nwane, Abraham Fomena, Josiane Etang

**Affiliations:** 1Laboratory of Animal Biology and Physiology, Faculty of Sciences, University of Yaoundé I, P.O. Box 337 Yaoundé, Cameroon; mbalira@yahoo.fr (L.R.M.); mandengelysee@yahoo.fr (S.E.M.); abfomena@yahoo.fr (A.F.); 2Institut de Recherche de Yaoundé (IRY), Organisation de Coordination pour la lutte contre les Endémies en Afrique Centrale (OCEAC), B.P. 288 Yaoundé, Cameroun; hpaawono@yahoo.fr (P.H.A.-A.); ewolfgang388@gmail.com (W.E.E.); antonio_nk@yahoo.fr (C.A.-N.); jctotofr@yahoo.fr (J.-C.T.); philino07@yahoo.fr (P.N.); 3Laboratory of Animal Biology and Physiology, University of Douala, P.O. Box 24157 Douala, Cameroon; 4Laboratoire d’Ingénierie Mathématique et Systèmes d’Information, Ecole Nationale Supérieure de Polytechnique (ENSP), Université de Yaoundé I, B.P. 337 Yaoundé, Cameroun; fesuhbe@yahoo.co.uk; 5Centre de Recherche sur les Filarioses et Maladies Tropicales (CRFilMT), B.P. 5797 Yaoundé, Cameroun; 6Faculty of Medicine and Pharmaceutical Sciences, University of Douala, P.O. Box 2701 Douala, Cameroon; 7Institute for Insect Biotechnology, Justus-Liebig University Gießen, Winchesterstr. 2, 35394 Gießen, Germany

**Keywords:** malaria vectors, *Plasmodium* transmission, dam construction, Memve’ele, Cameroon

## Abstract

Dam constructions are considered a great concern for public health. The current study aimed to investigate malaria transmission in the Nyabessan village around the Memve’ele dam in South Cameroon. Adult mosquitoes were captured by human landing catches in Nyabessan before and during dam construction in 2000–2006 and 2014–2016 respectively, as well as in the Olama village, which was selected as a control. Malaria vectors were morphologically identified and analyzed for *Plasmodium falciparum* circumsporozoite protein detection and molecular identification of *Anopheles (A.) gambiae* species. Overall, ten malaria vector species were identified among 12,189 *Anopheles* specimens from Nyabessan (*N* = 6127) and Olama (*N* = 6062), including *A. gambiae* Giles (1902), *A. coluzzii* Coetzee (2013), *A. moucheti* Evans (1925), *A. ovengensis* Awono (2004), *A. nili* Theobald (1903), *A. paludis* Theobald (1900), *A. zieanni*, *A. marshallii* Theobald (1903), *A. coustani* Laveran (1900), and *A. obscurus* Grünberg (1905). In Nyabessan, *A. moucheti* and *A. ovengensis* were the main vector species before dam construction (16–50 bites/person/night-b/p/n, 0.26–0.71 infective bites/person/night-ib/p/n) that experienced a reduction of their role in disease transmission in 2016 (3–35 b/p/n, 0–0.5 ib/p/n) (*p* < 0.005). By contrast, the role of *A. gambiae s.l.* and *A. paludis* increased (11–38 b/p/n, 0.75–1.2 ib/p/n) (*p* < 0.01). In Olama, *A. moucheti* remained the main malaria vector species throughout the study period (*p* = 0.5). These findings highlight the need for a strong vector-borne disease surveillance and control system around the Memve’ele dam.

## 1. Background

Each environmental change, whether occurring as a natural phenomenon or through human intervention, may alter the ecological balance within which disease hosts, vectors, and parasites develop [1,2]. Among the major environmental changes are deforestation, urbanization, agriculture, water drainage, and climate change, which affect the lifecycles and anthropogenic interactions between insect vectors of pathogens and human populations, with significant impact on disease transmission.

In many countries of sub-Saharan Africa, hydraulic infrastructures are widely advocated as a crucial investment for economic growth and reduction of poverty [3,4]. However, building these infrastructures imposes huge financial costs and social constraints that may indirectly affect human and animal health. For instance, the construction of dams results in physical, chemical, and biological changes in natural ecosystems, including disturbances of the targeted hydrological system and riverine ecosystem (soil, vegetation, and microclimate) [5]. These changes may lead to the proliferation of insect vectors [6,7,8,9,10,11] and increase the transmission of vector-borne and water-related diseases [12]. Vector-borne diseases afflicting populations surrounding dams include malaria [13], schistosomiasis [14], onchocerciasis, filariasis, and dracunculosis [5,15]. Among these, malaria is a major public health problem with 219 million cases and 435,000 deaths in the world in 2017, which includes 90% of cases in sub-Saharan Africa [16]. In the equatorial forest region, malaria is transmitted by *Anopheles gambiae*, *A. coluzzii*, *A. funestus*, *A.nili*, and *A. moucheti* [17,18]. The high adaptability of the most efficient malaria vector species [19], bottlenecks in control program implementation, financial constraints and poor community compliance to control measures, rapid increase of insecticide resistance in targeted vectors, limitations in the effectiveness of currently available tools (long-lasting insecticidal nets, indoor residual spraying, larvicidal treatment) especially concerning outdoor and residual malaria transmission, and the challenges in developing new vector control tools [20] constitute serious threats to the reduction of malaria within areas under water development projects (WDP) [21]. Meanwhile, these infrastructure projects aim to provide hydroelectricity, water for irrigation, domestic and/or industrial use, or the containment of flooding. According to the International Committee on Large Dams (ICOLD) [22], a large dam is characterized by walls higher than 15 m (or in some cases between 10 and 15 m high), a crest length of over 500 m, a storage capacity of more than 1,000,000 m^3^, and a spilling capacity of 2000 m^3^. Conversely, most dams that are less than 15 m high are considered small dams [23]. In 2005, it was estimated that a total 3.1 million people were at risk of malaria due to dams in sub-Saharan Africa [24]. Sanchez et al. [25] reported different patterns of malaria transmission in areas around large and small dams. Using a more extensive data set, Kibret et al. [26] reported that more than 15 million people were at risk and that 1.1 million cases a year were associated with 1268 large dams in South-Saharan Africa. The need for a thorough investigation on the impact of dam construction on human health has been highlighted in the framework of environmental change and human disease prevention [27,28,29]. However, specific effects on malaria transmission in the context of climate and demographic changes are poorly understood and seldom investigated [30]. As water storage provides potential mosquito breeding habitats and climate determines the development of mosquitoes and malaria parasites, it is important to understand how their interaction with dam construction affects malaria transmission [30].

In Cameroon, a study carried out in 1979 around the Bamendjin large dam (17 m height) across the Sanaga River, revealed that malaria transmission in this area was high and occurred throughout the year due to stagnant water favoring the breeding of *A. funestus* [31]. However, no information on the situation of malaria transmission around the Memve’ele hydroelectric dam (20 m height, 80 million m^3^ water storage capacity), which has been under construction since June 2012, is available. In order to advise the National Malaria Control Programme (NMCP) on the malaria situation around dams and other major man-made ecological transformations, the current study aimed at assessing the evolution of malaria entomological indicators around the Memve’ele dam in South Cameroon, where malaria transmission is perennial.

## 2. Methods

### 2.1. Study Design

The study combined retrospective and prospective malaria entomological surveys. Retrospective data were collected from the results of entomological surveys conducted from 2000 to 2006 and stored in a mosquito database at the Malaria Research Laboratory of Organisation de Coordination pour la lute contre les Endémies en Afrique Centrale (OCEAC). Prospective data were collected through cross-sectional surveys conducted in Nyabessan and Olama villages from 2014 to 2016 (July–December), two to four years after the launch of the Memve’ele dam construction in 2012.

### 2.2. Study Sites

The study was carried out in Nyabessan (2°80′ N, 10°25′ E) and Olama (3°24′ N, 11°18′ E), two villages situated in the equatorial forest in Cameroon (Figure 1).

Nyabessan is located on the edge of the Memve’ele dam across the Ntem River, at 220 km from Yaoundé, the capital city of Cameroon. The local human population has increased from ≈200 inhabitants in 2012 to ≈600 inhabitants in 2016, following the influx of workers involved in the dam construction. Most of the dwellings are made of temporary materials, but some houses are built with cement material. The area is characterized by the equatorial climate, with two rainy seasons from March–June and from September–November, and two dry seasons from December–February and from July–August. The average annual rainfall is 1500 mm and the average annual temperature is ≈25 °C. In 2000 and 2001, mosquito samples were collected in Oveng (2°10′ N, 10°30′ E), a village situated in the Nyabessan neighborhood, 5 km upstream along the Ntem river, and sheltering less than 100 inhabitants.

Olama, located on the edge of the Nyong River, at 150 km from the Memve’ele dam, was selected as the control village with no nearby hydraulic infrastructures. This village shelters ≈300 inhabitants and the houses are mostly built with clay. Similar to Nyabessan, this area is characterized by the equatorial climate.

### 2.3. Mosquito Collection and Morphological Identification

Adult mosquitoes were collected by human landing catches (HLC) performed by volunteers living in the prospected study sites. We used a stratified random sampling approach to select the houses and volunteers for mosquito collection. Each village was divided into three to five collection units, and one house was randomly selected in each unit for mosquito collection. At each selected house, one volunteer collected mosquitoes inside the house whereas another one collected mosquitoes from outside the house. In each village, mosquito collection was performed per night by 12–20 volunteers working in two teams of 6–10 collectors. The first team collected from 19:00 to 01:00 and the second team collected from 01:00 to 05:00. To avoid bias due to the attractiveness and skills of the mosquito collectors, the volunteers rotated between collecting indoors and outdoors every two hours. Mosquitoes were collected through cross-sectional entomological surveys conducted during small rainy seasons (Mars, April, May), small dry seasons (June, July), big rainy seasons (August, September), and big dry seasons (December). The total number of person–nights for mosquito collection in each study sites was 109 in Nyabessan and 106 in Olama (Table 1). Collected mosquitoes were morphologically identified using the standard identification keys [32,33], and individually stored in labeled 1.5 mL Eppendorf tubes containing silica gel desiccant for further analysis.

### 2.4. Molecular Identification

All female mosquitoes morphologically identified as belonging to the *A. gambiae* complex were subjected to molecular identification. DNA was extracted from the legs and wings using CTAB (Cetyl trimethyl ammonium bromide) 2% as described by Collins et al. [34] and used for identification of sibling species by means of the SINE200 (short interspersed elements) polymerase chain reaction (PCR) protocol as described by Santolamazza et al. [35]. SINEs are homoplasy-free and co-dominant genetic markers for differentiation of *A. gambiae* and *A. coluzzii*.

### 2.5. Determination of Plasmodium Falciparum Infectivity

For each female anopheline mosquito, the head and thorax were tested for the presence of circumsporozoite protein (CSP) of *Plasmodium falciparum* using enzyme-linked immunosorbent assay (ELISA) as described by Burkot et al. [36] and later modified by Wirtz et al. [37].

### 2.6. Statistical Analysis

For both retrospective and prospective data, the following entomological indicators were calculated:The Human Biting Rate (HBR) which represents the average number of bites received per person per night;The Infection Rate (IR) that measured the proportion of mosquitoes positive for *P. falciparum* Circumsporozoite antigene;The Entomological Inoculation Rate (EIR), which is the number of infective bites received per person per night; EIR is calculated as the product of the HBR and IR.

The 2-sample test for equality of proportions was used to compare the proportions of each species before and after the launch of the dam construction and the ANOVA to compare the average HBR, IR, and EIR at 95% confidence interval. Data were analyzed using the R 3.5.0 software (R Development Core Team, Vienna, Austria, 2018).

### 2.7. Ethics Approval and Consent to Participate

The study was conducted under the ethical clearance no. 2016/01/685/CE/CNERSH/SP delivered by the Cameroon National Ethics (CNE) Committee for Research on Human Health. All volunteers participating in human landing catches signed a written informed consent form indicating their willingness to take part in the study. They also received free malaria prophylaxis.

## 3. Results

### 3.1. Composition and Abundance of Anopheline Fauna in Nyabessan and Olama

A total of 12,189 female anophelines were collected in both study sites from 2000–2016. The distribution of species in the two study sites is shown in Table 1.

In Nyabessan, 6127 specimens belonging to eight species/species complexes were caught using 53 person–nights before the Memve’ele dam construction (MDC) (2000–2006) and 56 person–nights during MDC (2014–2016). The same species were identified either before MDC or during MDC, although at significantly different proportions (*p* < 0.0002), except *A. obscurus* which remained at a very low proportion (less than 0.05%) (*p* = 0.8) (Table 1). Before MDC, the most abundant species was *A. ovengensis* (53%) and *A. moucheti* (39%) followed by *A. gambiae s.l.* (5%) and *A. marshallii* (~3%) (Figure 2). *A. ovengensis* is one of the malaria vector species belonging to the *A. nili* group. Until now, this species has exclusively been found in the Nyabessan area in South Cameroon. Over time, *A. paludis*, which was scarce before MDC, became one of the major malaria vector species, representing around 22% of the total number of anophelines caught during MDC (*p* < 0.0001) (Table 1). The species *A. obscurus* remained scarce throughout the study period (*p* = 0.8).

Over time, the proportions of *A. moucheti*, *A. ovengensis*, and *A. marshallii* significantly decreased (*p* < 0.0001), while the proportions of *A. gambiae s.l.* increased from 5% before MDC to 15% after MDC (*p* < 0.0001). PCR analysis of 164 specimens of *A. gambiae* complex collected during three consecutive years (2014–2016) revealed changes in the proportions of two sibling species, from 93% *A. gambiae s.s.* and 7% *A. coluzzii* in 2014 to 61% *A. gambiae s.s.* and 39% *A. coluzzii* in 2016. Similar to *A. coluzzii*, which was nearly absent before 2006 (all the 37 specimens tested as *A. gambiae*), *A. ziemanni* and *A. coustani* occurred during the implementation of activities associated with dam construction, although at low frequencies (<3%).

In Olama, 6062 anophelines were caught by 106 person–nights (PSN) (70 PSN before MDC, 36 PSN after MDC). The composition of anopheline fauna in Olama was similar to Nyabessan, with 8 vector species/groups of species found in the two study sites, including *A. nili* (Table 1). However, *A. moucheti* remained the most abundant malaria vector species (69%–99%) regardless of the collection period, although its proportion significantly decreased in 2014 (69%) (*p* < 0.0001) and was balanced with *A. ziemanni* (29%) (Figure 2). The proportion of each other species including *A. gambiae s.l.* remained less than 3% during the seven entomological surveys conducted from 2000–2016 (*p* > 0.05).

### 3.2. Night Biting Rates and Biting Cycles of Malaria Vector Populations, Indoors and Outdoors in Nyabessan and Olama

#### 3.2.1. Night Biting Rates Indoors and Outdoors

For each of the six major vector species (*A. gambiae s.l.*, *A. marshallii*, *A. moucheti*, *A. ovengensis*, *A. paludis*, and *A. ziemanni*) that were identified, the average in/outdoor HBR before and during MDM in Nyabessan versus Olama are presented in Figure 3 and the effect plot in Figure 4. The ANOVA table for the linear model of HBR (Table 2) revealed that the variables period, species, interaction between period and species, hours, interactions between period and position (in/out), species and position, location and hour significantly explained variations in HBR. Q-Q plot of standardized residuals versus theoretical quantiles for this model revealed a slight deviation of the residuals from normal, with quite heavy tails.

The ANOVA model of human biting rates as a function of different time slots and species is provided in Table 3.

In Nyabessan, the mean HBRs of *A. gambiae* were 3–6 bites/person/night before and during MDC (Figure 3). For *A. moucheti* and *A. ovengensis*, the HBRs decreased from 25 bites/person/hour and 40 bites/person/hour before MDC to 18 bites/person/hour and 15 bites/person/hour after MDC respectively (*p* < 0.001). The species *A. paludis*, which was scarce before MDC and after MDC until 2015, became one of the major species in 2016 with up to 35 bites/person/night (*p* = 0.003). Conversely, *A. marshallii* was found before MDM (0.3–4.4 bites/person/night), but became scarce later on after MDC (only 0.4 bite/person/night in 2014), although the difference was not significant (*p* = 0.72). The species *A. ziemanni* was only found after MDC, although at a very low density, less than 1 bite/person/night.

In Olama, the mean HBR of *A. moucheti* was more than three times higher before MDC (84 bites/person/night respectively) than later on during MDC (23 bites/man/night from) (Figure 3). Nevertheless, the HBR of this species was still high compared with that of the five remaining vectors. The average HBRs of the remaining vectors (e.g., *A. paludis* and *A. marshallii*) were frequently less than 1 bite/person/night (Figure 3), except *A. ziemanni*, which reached 6 bites/person/night in 2014.

#### 3.2.2. Indoor and Outdoor Biting Cycles

The effect plot showing the mean HBRs outdoors/indoors in Olama and Nyabessan is given in Figure 5. The average indoor/outdoor HBR of the six major vectors was not significantly different before MDC (60 bites/person/night) versus during MDC (52 bites/person/night) (*p* = 0.48) (Table 2). However, there was a significant difference of HBR between outdoor and indoor collections at species level (*p* = 0.04), especially in *A. moucheti*, *A*. *ovengensis*, and *A. paludis* as compared with *A. gambiae* (*p* < 0.004) (Table 3). The HBR was not significantly different between Nyabessan and Olama (*p* = 0.1) (Table 2).

Throughout the nights from 19:00 to 05:00, variable trends of biting activity were noted either in Nyabessan (Figure 5) or in Olama (Figure 6). The effect plot showing the estimated mean HBRs indoors versus outdoors for each species is presented in Figure 7. The biting cycles varied according to the survey period (before/after MDC), the biting position (indoor/outdoor), and the anopheline species. In general, the biting rates were significantly higher between 23:00 and 01:00 or between 01:00 and 03:00 (*p* < 0.02) as compared with 19:00 and 21:00 (Table 3). Four major vector species (*A. gambiae*, *A. moucheti*, *A. ovengensis*, and *A. paludis*) showed an ability to indifferently bite indoors and outdoors throughout the study period (*p* = 0.48) (Table 2).

In Nyabessan, *A. gambiae* hourly HBRs before MDC were flat either outdoors or indoors. However, during MDC, the peaks of biting activity occurred between 01:00 and 03:00 indoors, or after 05:00 outdoors (4 bites/person/hour). With *A. moucheti*, the biting cycle indoors or outdoors displayed similar tendencies before and during MDC, with peaks of hourly HBRs between 23:00 and 01:00 indoors or earlier (21:00 and 23:00) outdoors. Furthermore, a significant decrease of HBRs was clearly observed from 5–15 bites/person/hour in/outdoors before MDC to less than 5 bites/person/hour during MDC, no matter the position of mosquito collection (Figure 5 and Figure 7). With *A. ovengensis*, variable shapes of biting cycles were observed. Before MDC, the HBRs gradually decreased from 10 to 15 bites/person/hour (indoors and outdoors) during the first half of the night to less than 5 bites/person/hour (indoors and outdoors) during the second half of the night (Figure 5 and Figure 7). During MDC, the HBRs remained less than 5 bites/person/hour during the whole night. More importantly, we have found *A. paludis* with a high endophagic tendency, and a peak of HBR between 03:00 and 05:00 (up to 10 bites/person/night). The biting cycle of this species outdoors was flat.

In Olama, *A. moucheti* displayed high HBRs indoors before MDC (up to 25 bites/person/night) versus outdoors (up to 15 bites/person/night), with a peak of activity between 23:00 and 01:00 as in Nyabessan. The peak biting activity outdoors at this period was recorded between 01:00 and 03:00. During MDC, the HBRs was less than 7 bites/person/night throughout the nights, either indoors or outdoors. The biting cycles of the remaining anopheline species could not be analyzed due to their absence or very low densities.

### 3.3. Infection Rates of Malaria Vectors in Nyabessan and Olama

Data on *P. falciparum* infection rates (IR) by vector species in Nyabessan and Olama are presented in Figure 8 and the ANOVA model in Table 4.

In Nyabessan, four anopheline species, namely *A. gambiae s.l.*, *A. marshallii*, *A. moucheti*, and *A. ovengensis* were mainly responsible for *Plasmodium* transmission irrespective of the survey year, followed by *A. paludis*, which arose as the local malaria vector species in 2016. No significant difference of IR was observed between the period before (1.1%) and during dam construction (1.0%) (*p* = 0.16). In 2015, the IR for all five species was 0.0%. Some variations of IR were observed among the vector species, although theses variations were not significant (*p* = 0.07) (Figure 8). None of the variables period, species, location, and their interaction significantly explained variations in IR (Table 4, Figure 8). Q-Q plot of standardized residuals versus theoretical quantiles for this model revealed slight deviation of the residuals from normal, with quite heavy tails.

No significant difference of IR was observed between Olama and Nyabessan (*p* = 0.68) (Table 4). However, in Olama, malaria was mainly transmitted by *A. moucheti* (0.7% to 2.5% IR) throughout the study period, followed by *A. gambiae s.l.* The IR of *A. gambiae s.l.* decreased from 6% before MDC to 0.0% during MDC, but this decrease was not significant as shown on the effect plot (Figure 8). For the remaining vector species, the IR was almost less than 1.5%.

### 3.4. Entomological Inoculation Rates of Malaria Vectors in Nyabessan and Olama

Data on entomological inoculation rates (EIR) by vector species in Nyabessan and Olama are presented in Figure 9 respectively and the ANOVA model in Table 5.

In Nyabessan, the entomological inoculation rate for each of the five main vector species was mostly less than 1 infective bite/person/night (ib/p/n), with no significant difference between the two study periods (*p* = 0.06). None of the variables period, species, location, and their interaction significantly explained variations in EIR (Table 5). Q-Q plot of standardized residuals versus theoretical quantiles for this model revealed very little deviation of the residuals from normal. However, the EIR of *A. paludis* significantly increased from 0.0 ib/p/n before MDC to 1.2 ib/p/n during MDC (Figure 9). The EIRs of each of the remaining species were less than 1 ib/p/n throughout the two study periods.

Similar to the IR, the overall EIR for all five species was not significantly different between Olama and Nyabessan (*p* = 0.06). The highest EIRs in Olama were recorded in *A. moucheti* before MDC (≈1.5 ib/p/n), afterward it decreased to less than 1.0 ib/p/n and remained at that level until 2016. For the remaining vector species, the EIR was almost 0.0 ib/p/n.

## 4. Discussion

The data presented in this study were collected during seven years since year 2000, 12 years before the launch of the Memve’ele dam construction in 2012, then for 3 years during the construction until the impoundment of the dam in 2016. Although entomological surveys were not conducted for consecutive years or during the entirety of the dry and rainy seasons, the current study brings up-to-date information on the variable roles of *An. moucheti* together with other anopheline vector species in the evolution of malaria transmission, in a dual context of intensive environmental modifications and Long Lasting Insecticidal Net (LLIN) interventions in Nyabessan versus Olama, and similar localities in the equatorial regions of Cameroon.

Of the forty malaria vector species previously identified in Cameroon [38,39], ten species were recorded, among which *A. gambiae*, *A. coluzzii*, *A. moucheti*, and *A. nili* known as the major malaria vectors in the equatorial forest region, alongside *A. funestus*, which was not recorded either in Nyabessan or in Olama. Each *Anopheles* species was characterized by typical habitat preferences, including exposure to sunlight, turbidity of the water, and presence of vegetation, pH or nitrate and phosphate concentrations of the water [40]. These environmental factors are specific for each dam and its shoreline. Furthermore, with the presence of a dam, flooding is common upstream from the dam. During flooding, water covers a vast area of land and when the water recedes, little ponds are left within the flooded area which are suitable breeding sites for mosquitoes. While fluctuations were observed in the distribution of four major vector species in Nyabessan before and during dam construction across the Ntem River, *A. moucheti* remained the major malaria vector species in Olama during the whole study period, followed by *A. gambiae*. Either in Nyabessan or in Olama, malaria vector biting rates were highly variable between indoors and outdoors depending on the period and species. In Nyabessan, the most interesting trend was that of *A. paludis*, which in 2016 when its HBR indoors was higher when compared to other malaria vector species as well as with its HBR outdoors. By contrast, *A. ovengensis* showed an exophilic tendency. These findings emphasize the variability of malaria vectors’ biting behavior depending on the environmental factors and the high risk of *Plasmodium* transmission indoors and outdoors as well, as previously stressed [41,42]. Improved socio-economic variables, effective vector control programs or changes in health-seeking behavior are also pivotal in determining whether a water project triggers an increase in malaria transmission [43]. From the time when the first nationwide free distribution of LLINs was conducted in 2011–2012 and the second one in 2015, the general use of LLINs for malaria prevention might lead to temporary changes in vector biting behavior and malaria transmission system. The reduction of man-vector contact and increase of outdoor biting rates underline the need for outdoor vector control measures to complement LLNs interventions in these areas.

The increase of malaria transmission intensity in 2016 in Nyabessan was led by *A. gambiae* and *A. paludis* in the progressive replacement of *A. moucheti*. This switch of vector species distribution is consistent with previous studies conducted in some areas of the equatorial region of Cameroon that encountered considerable environmental modifications [17,44,45,46]. In contrast to Olama where no significant ecological change occurred during the last decade, malaria was permanently transmitted by *A. moucheti*, as previously reported by Antonio-Nkondjio et al. [42]. *A. moucheti* is widely distributed across West and Central Africa, playing a major role in malaria transmission in villages located close to rivers or slow-moving streams in the equatorial region [17,45,46,47]. The larvae of this species are frequently associated with lentic rivers, low temperatures, and the abundance of aquatic vegetation at the edge of the river [48]. Increased urbanization and deforestation as well as lower-scale landscape modification such as river banks cleaning for gardening and/or recreational purposes were shown to be highly detrimental to this species, fostering changes in the malaria vector system composition with a higher prevalence of *An. gambiae*, taking the lead over *An. moucheti* [48].

*Anopheles paludis* is known as a secondary malaria vector, essentially found in the equatorial forest and savanna from West Africa to East Africa [49]. In Cameroon, this anopheline species has been recorded in 21 villages [50], its role in malaria transmission in this country has been confirmed since 1964 [51]. However, very little was known about the increase of its frequency and its important role in malaria transmission in Nyabessan. The common breeding sites of this mosquito species are expanses of clear water with aquatic or semi-aquatic vegetation, most often sheltered from the forest canopy. Larvae are found in pools, ponds, backwaters, upright vegetation, and even along grassy banks of rivers. Previous studies revealed that *A. paludis* is very anthropophilic, exophilic, and exophagous [52]. Its biting cycle showed either an activity distributed over the night, or a peak of aggressiveness at dusk. This *Anopheles* species was reported to bite in the undergrowth during the day in Cameroon and the Central African Republic [53]. The current study provided up-to-date information on the important and increasing role of *A. paludis* in malaria transmission in Nyabessan among the vectors that were previously predominant in the study area such as *A. moucheti* and *A. ovengensis* as well as those that were scarce, such as *A. gambiae s.l*. The high frequency among the collected anopheline samples, the infection rate, biting cycles outdoors and indoors as well as the EIR reported here reveal that *A. pludis* may be responsible for high malaria risk in Nyabessan during the coming years. These data are comparable with those reported by Kharch et al. [52] in the Bandundu region of the Democratic Republic of Congo during the dry season. The peak biting activity of *A. paludis* in Nyabessan occurred between 03:00 and 05:00, suggesting significant daytime biting rates. Although such biting behavior is consistent with previous reports [53], the daytime biting behavior may also be related to high coverage of LLINs in households. The risk of malaria transmission might therefore become very high when people are awake and outside LLINs.

Furthermore, the current study revealed a peak of *A. gambiae s.l.* biting activity between 03:00 and 05:00. Together with *A. paludis*, *A. gambiae* was found to increasingly occur in Nyabessan with densities rising over time, with the highest value obtained in 2016. Conversely, the densities of *A. gambiae* in Olama remained low and constant throughout the collection period. Environmental modifications for habitat construction led by the massive arrival of dam workers in Nyabessan might have created suitable breeding sites for *A. gambiae*. This species is known to colonize aquatic habitats resulting from housing and agricultural activities in peri-urban areas in sub-Saharan Africa [42,54,55,56,57]. Molecular identification of specimens belonging to the *A. gambiae* complex revealed that samples collected in Nyabessan were composed of two species, namely *A. gambiae s.s.* (76.3%) and *A. colluzzii* (1.43%), while only *A. gambiae s.s*. was previously found in this locality [58]. Interestingly, the few specimens collected in Olama from 2014–2016 (*N* = 7) were identified as *A. gambiae s.s.* The apparently new occurrence of *A. colluzzi* in Nyabessan may be associated with anthropization of the environment, opportunistic behavior of this species, as well as haphazard and uncontrolled human activities [59,60]. Furthermore, *A. paludis*, which was scarce before dam construction in Nyambessan, became wide spread in 2016, while in Olama, its proportion during the same period remained very low (0.43%). However, seasonal variations observed in the density of this species may also be related to rainfall or stream flow [38].

Lastly, the intermittent presence of *A. ziemanni* in at both study sites at low densities and the *Plasmodium* infection rates in Olama highlights its secondary role in local malaria transmission. Previous studies revealed *A. ziemanni* as a major malaria vector in Northern Cameroon [61]. This species typically breeds in swampy areas and large water bodies, such as those resulting from environmental modifications around the Ntem River for the Memve’ele dam construction.

## 5. Conclusions

Although the composition of anopheline fauna was similar in Nyabessan and Olama, variations observed in malaria transmission patterns (changes in vector species distribution, increased *A. paludis* abundance and entomological inoculation rates) following large dam construction in Nyabessan may lead to an increase in malaria cases. The data reported here are of a great importance for the implementation of additional measures to sustain malaria prevention in Nyabessan. The peak of biting activity between 03:00 and 05:00 and outdoor biting rates underlines the need for complementary vector control measures, especially those targeting outdoor malaria transmission. Further entomological surveys are needed to monitor the malaria entomological profile following the impoundment of the Memve’ele dam. Furthermore, future water resource developments should include a comprehensive assessment of potential health impacts and a multifaceted action plan for health improvement.

## Figures and Tables

**Figure 1 ijerph-16-01618-f001:**
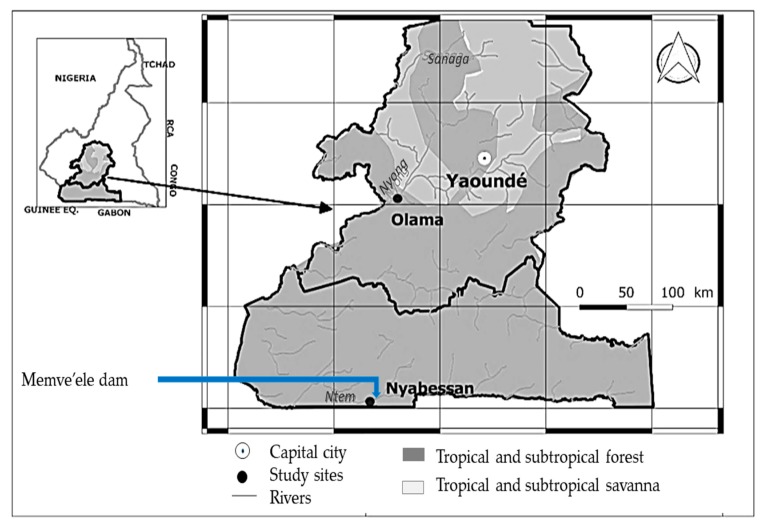
A map of the center and southern regions of Cameroon that show the study sites.

**Figure 2 ijerph-16-01618-f002:**
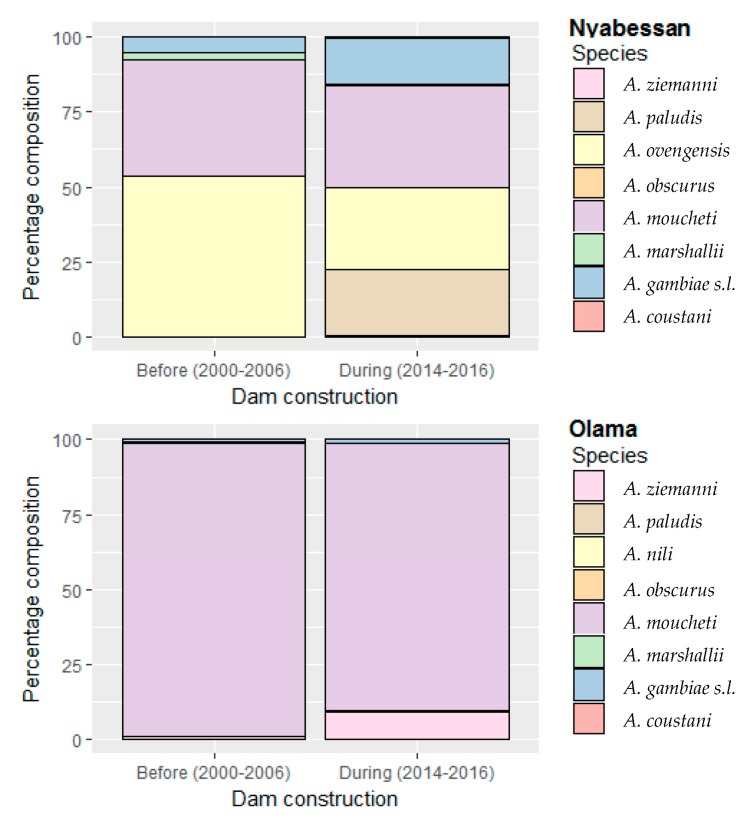
Stacked bar plots showing the percentage composition of anopheline fauna throughout time points in Nyabessan and Olama, before versus during the Memve’ele dam construction.

**Figure 3 ijerph-16-01618-f003:**
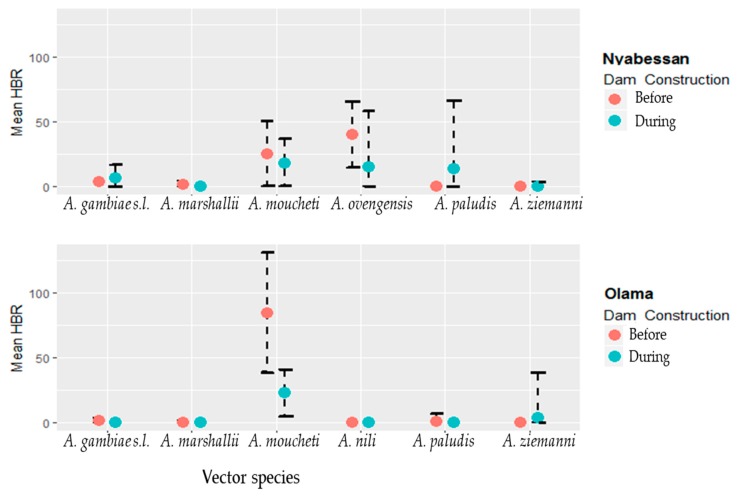
Mean night human biting rates (HBR) of six major anopheline species in Nyabessan and Olama before and during dam construction.

**Figure 4 ijerph-16-01618-f004:**
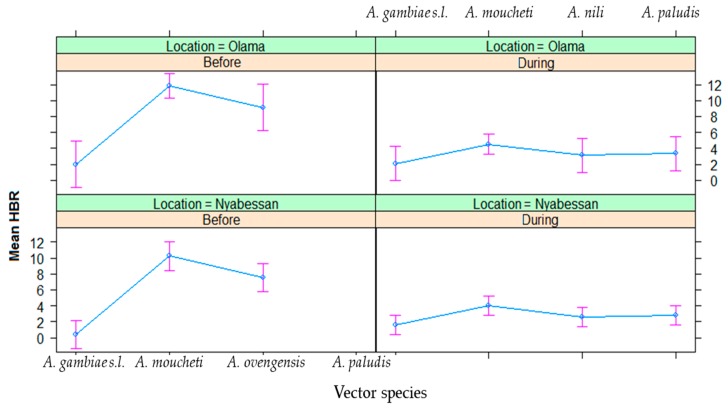
Effect plot showing the estimated mean HBRs in Olama and Nyabessan before and after dam construction.

**Figure 5 ijerph-16-01618-f005:**
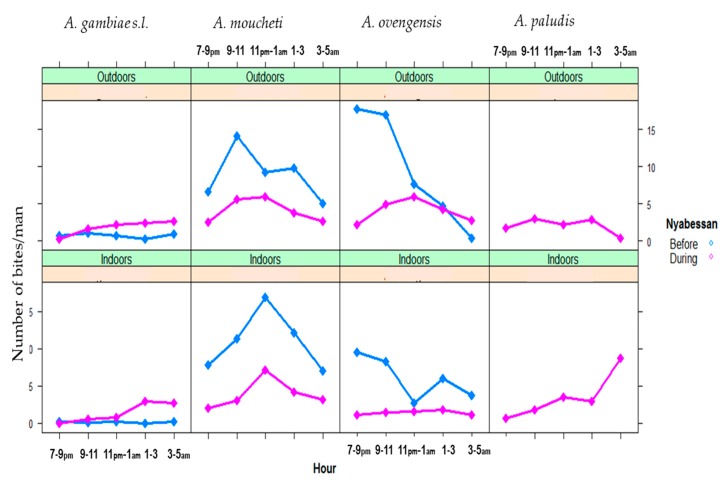
Patterns of the hourly biting curves for four major malaria vector species in Nyabessan, indoors and outdoors before and during dam construction.

**Figure 6 ijerph-16-01618-f006:**
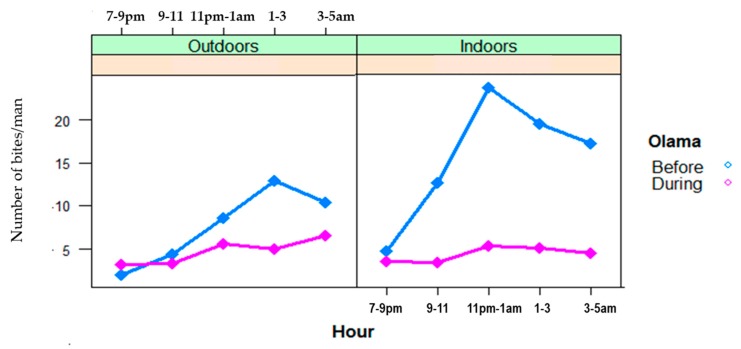
Patterns of *A. moucheti* hourly biting curves in Olama.

**Figure 7 ijerph-16-01618-f007:**
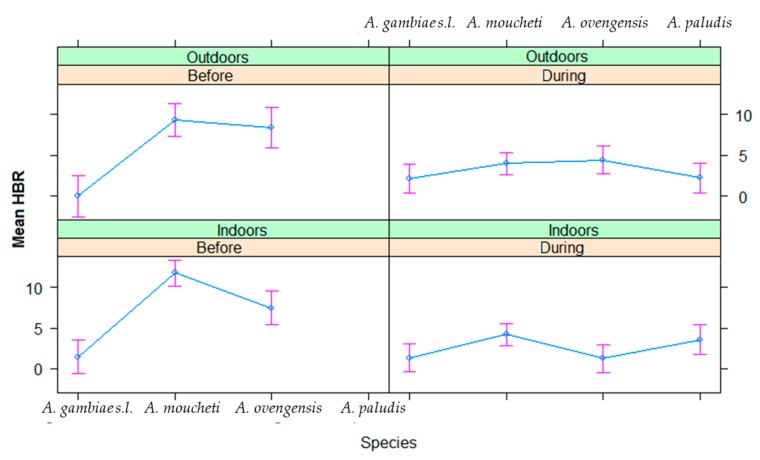
Effect plot showing the estimated mean HBRs in outdoors versus indoors in Nyabessan before and after dam construction.

**Figure 8 ijerph-16-01618-f008:**
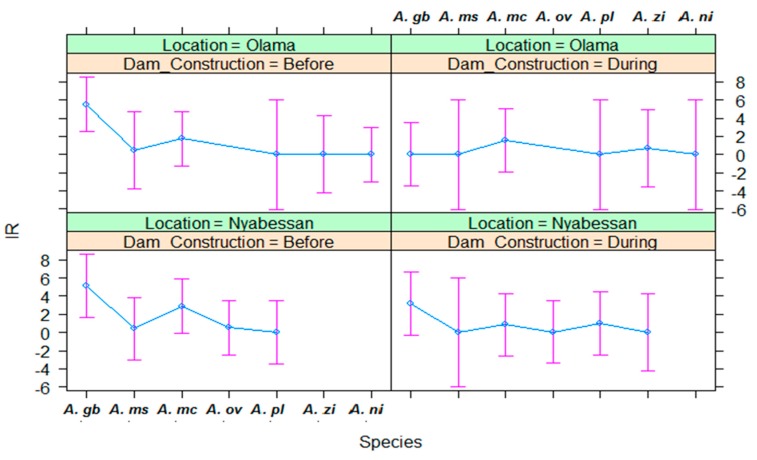
Effect plot showing the estimated mean IRs in Olama and Nyabessan before and after dam construction. IR: infection rate; *A. gb*: *Anopheles gambiae s.l.*; *A. ms*: *Anopheles marshallii*; *A. mc*: *Anopheles moucheti*; *A. ov/ni*: *Anopheles ovengensis or A. nili*; *A. pl*: *Anopheles paludis*; *A. zi*: *Anopheles ziemanni*.

**Figure 9 ijerph-16-01618-f009:**
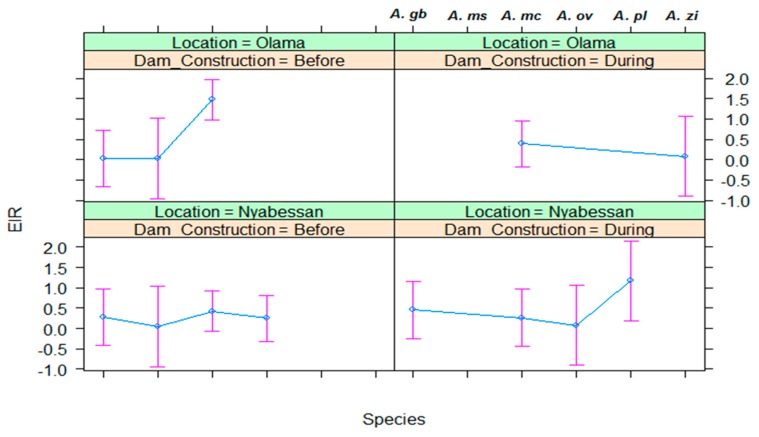
Effect plot showing the estimated mean EIR of malaria vectors in Nyabessan and Olama before versus during dam construction. *A. gb*: *Anopheles gambiae s.l.*; *A. ms*: *Anopheles marshallii*; *A. mc*: *Anopheles moucheti*; *A. ov/ni*: *Anopheles ovengenesis of A. nili*; *A. pl*: *Anopheles paludis*; *A. zi*: *Anopheles ziemanni*.

**Table 1 ijerph-16-01618-t001:** Distribution of malaria vector species among the *Anopheles* samples caught in Nyabessan and Olama.

Study Site	Species Distribution before Dam Construction	Species Distribution during Dam Construction	*p*-Value of 2-Sample Chi-Square Test for Equality of Proportions
	2000–2006	2014–2016	
**Nyabessan**			
Nb. pnc.	53	56	
*A. gambiae s.l.*	190 (5.1)	358 (15.0)	<0.0001
*A. moucheti*	1455 (38.9)	817 (34.2)	0.0002
*A. ovengensis*	1992 (53.3)	651 (27.2)	<0.0001
*A. paludis*	4 (0.1)	524 (21.9)	<0.0001
*A. ziemanni*	0	11 (0.5)	0.0001
*A. marshallii*	97 (2.6)	14 (0.6)	<0.0001
*A. coustani*	0	13 (0.5)	<0.0001
*A. obscurus*	0	1 (0.04)	0.8214
*Total*	3738 (100)	2389 (100)	
**Olama**			
Nb. pnc	70	36	
*A. gambiae s.l.*	45 (0.9)	10 (1.1)	0.6503
*A. moucheti*	5037 (97.9)	814 (89.0)	<0.0001
*A. nili*	10 (0.2)	1 (0.1)	0.8925
*A. paludis*	5 (0.1)	4 (0.4)	0.0460
*A. ziemanni*	41 (0.8)	83 (9.1)	<0.0001
*A. marshallii*	9 (0.2)	3 (0.3)	0.5783
*Total*	5147 (100)	915 (100)	

Nb. pnc.: Total number of person–night collection per site; Numbers in parentheses denote the proportion of each *Anopheles* species in relation to the total number of specimens caught.

**Table 2 ijerph-16-01618-t002:** ANOVA table for linear model of human biting rates as a function of periods before and during dam construction, location, species and biting position (indoor or outdoor), with their interactions.

Variable	Sum Sq	Df	F Value	Pr (>F)
Period	807.14	1	65.99	<0.0000 *
Location	29.12	1	2.38	0.1254
Species	551.60	3	15.03	<0.0000 *
Position	5.96	1	0.49	0.4864
Hour	182.87	4	3.74	0.0066 *
Period:Location	11.02	1	0.90	0.3444
Period:Species	306.99	2	12.55	<0.0000 *
Period:Position	77.74	1	6.36	0.0130 *
Location:Position	18.66	1	1.53	0.2191
Species:Position	103.11	3	2.81	0.0422 *
Period:Hour	36.34	4	0.74	0.5645
Location:Hour	121.40	4	2.48	0.0472 *
Species:Hour	197.81	12	1.35	0.2003
Position:Hour	57.92	4	1.18	0.3212
Period:Location:Position	47.27	1	3.86	0.0515
Period:Species:Position	12.06	2	0.49	0.6119
Period:Location:Hour	101.69	4	2.08	0.0875
Period:Species:Hour	107.66	8	1.10	0.3675
Period:Position:Hour	54.23	4	1.11	0.3556
Location:Position:Hour	20.33	4	0.42	0.7972
Species:Position:Hour	97.86	12	0.67	0.7803
Period:Location:Position:Hour	12.25	4	0.25	0.9090
Period:Species:Position:Hour	42.31	8	0.43	0.8997

Sum Sq: Sum of Squared errors; Df: Degree of freedom; F value: F statistic value for each coefficient; Pr: *p*-value for the *F*-test on the model. * denotes significant interaction of variables at 5% level.

**Table 3 ijerph-16-01618-t003:** ANOVA model of human biting rates as a function of different time slots and species.

Factor	Estimate	Std. Error	*t* Value	*p*-Value
**Intercept**	2.5602	0.9717	2.635	0.0091 *
**Hour** (ref = 07–09 p.m.)				
09–11 p.m.	1.5566	0.9024	1.725	0.0860.
11 p.m.–01 a.m.	2.7241	0.9024	3.019	0.0029 *
01–03 a.m.	2.2428	0.9024	2.485	0.0137 *
03–05 a.m.	1.4813	0.9024	1.642	0.1022
**Species** (ref = *A. gambae s.l.*)				
*A. moucheti*	5.4189	0.7578	7.151	<0.0001 *
*A. ovengensis*	2.9990	0.8821	3.400	0.0008 *
*A. paludis*	3.0243	1.0094	2.996	0.0031 *
*A. marshallii*	−2.682	7.699	−0.348	0.7289

* denotes significant difference of hourly HBR.

**Table 4 ijerph-16-01618-t004:** ANOVA table for model of infection rates (IR) as a function of periods before and during dam construction, location, and species.

Variable	Sum Sq	Df	F Value	Pr (>F)
Period	17.48	1	1.9789	0.16785
Species	114.85	6	2.1667	0.06856
Location	2.46	1	0.2787	0.60073
Period:Species	37.20	6	0.7017	0.64996
Period:Location	1.04	1	0.1177	0.73350
Species:Location	4.66	4	0.1317	0.96978
Period:Species:Location	12.05	3	0.4545	0.71569

**Table 5 ijerph-16-01618-t005:** ANOVA table for the model of entomological inoculation rate (EIR) as a function of periods before and during dam construction, location, and species.

Variable	Sum Sq	Df	F Value	Pr (>F)
Dam_Construction	0.856	1	4.056	0.0636
Species	2.484	5	2.355	0.0948
Location	0.864	1	4.095	0.0625
Dam_Construction:Species	0.079	2	0.188	0.8311
Dam_Construction:Location	0.652	1	3.092	0.1005
Species:Location	1.301	2	3.083	0.0777

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
