# Peer review of "Malaria Transmission around the Memve’ele Hydroelectric Dam in South Cameroon: A Combined Retrospective and Prospective Study, 2000–2016"

_ijerph, 2019, doi:10.3390/ijerph16091618_

Round 1

Reviewer 1 Report

Dear Authors,

Thank you very much for the opportunity to review your manuscript. I believe you have an incredible dataset that has a great potential to demonstrate the effect of dams and impoundment on the ecology of mosquitoes that transmit malaria, and therefore malaria transmission and cases. However, I believe that you have not sufficiently utilized the incredible dataset you have. In general, I find the analysis in the manuscript descriptive, simply describing the number of mosquitoes collected and the proportion with evidence of prior infection at the two different locations at the different periods. This is not sufficient to make inferences, such as answering questions like “Is there a difference between the number of mosquitoes with prior infection between before and after the dam construction?”. One might say that the numbers seek for themselves. However, as detailed on the bottom of page 3, mosquitoes were collected in one house in each collection unit in each village at every surveillance event. Descriptive statistics, such as those presented here, can only summarize the situation in each of these respected houses. However, I would presume that these houses were selected as representative samples of the population of interest, which is all the houses in each collection unit in each village. If you would like to make claims relative to all houses, which it sounds like you do later in the manuscript, descriptive statistics such as presented here is insufficient, and you need inferential statistics, so hypothesis tests about the population of interest. This is because there is random variation in your variables of interest (such as the number of mosquitoes collected), and only hypothesis testing and estimation (inferential statistics) can distinguish between what are genuine (statistically significant) differences. These would be required to support the claims you are making in your manuscript about the effects of dam and impoundment on the ecology of mosquitoes and malaria transmission, even if the descriptive numbers seem to support your conclusion; those only relate to the specific houses where you collected the data. If you lack the expertise to test statistical hypothesis using your data, I would recommend consulting a statistician who could help you with that. In addition, I believe you could improve the quality of your presentation significantly by turning Table 1, Table 2 and Table 3 into figures, and combining Figure 2-5 into a single Figure, as well as the subplots of Figure 6 into a single plot. These figures were obviously made in Microsoft Excel which is not the right software to make publication-quality figures. Please use a more professional program (such as the freely available software R, or SigmaPlot or JMP) to recreate these figures. Please see my detailed comments below:

1.       In the Abstract, page 1, line 25, please do not leave a space between “12, 189”. Just have “12,189”. Same goes on line 26 to “6, 062”. These occur again later, so please check and correct throughout the manuscript.

2.       In the Abstract, page 1, line 28, you have “16-50 bites/man/night-b/p/n”. Do you mean “16-50 bites/person/night”? If yes, please use that. This occurs throughout the manuscript, please check and correct.

3.       In the Abstract, page 1, line 29-30, it sounds like both A moucheti and A. ovengensis, as well as A. gambiae s.l. and A. paludis experience a reduction in their role in disease transmission. This is not what is shown later. Please correct this sentence to reflect the results later on.

4.       On page 2, line 52, please add the year to which these statistics on malaria case numbers refer to.

5.       On page 2, line 57, please put a comma after “Meanwhile”.

6.       On page 2, line 60 and 61 seems to be missing “over” or “more than” when referring to the threshold in terms of storage capacity and spilling capacity for large dams.

7.       On page 2, line 62, which year does the estimate of the number of people at risk in terms of malaria due to dams refer to?

8.       On page 2, line 80, what is the correct spelling for the country? “Cameroon” or “Cameroun”?

9.       On Figure 1, could you please indicate the precise location of the dam?

10.   On page 3, on the bottom, you describe the protocol for mosquito collection. This seems to be a form of stratified random sampling. Do you agree? If yes, please state that.

11.   On page 4, line 127-129, you describe the ELISA method that you used to determine the infection status of mosquitoes with Plasmodium falciparum. Does this measure the current infection status of mosquitoes, or the infection history of these mosquitoes, since it measures antibodies? Or does it not matter because mosquitoes never clear the infection?

12.   On page 4, line 133, you talk about the statistical analysis, and the entomological indicators that you calculated. The first one you mention as “man biting rate (m.a.)”. However, elsewhere you refer to this as the Human Biting Rate (HBR). I believe this later is more widespread in the literature, so I suggest you to use that consistently. Please check and correct throughout.

13.   On page 4, line 139, you mention that you conducted a chi-square test on the number of mosquitoes collected. However, I wonder if that is the appropriate test, given that the number of some species is very low. A Fisher’s exact test might be more appropriate to test the null hypothesis that the proportion of each species is the same between the two periods (2000-2006) and (2014-2016).

14.   On page 4, line 141, you mention an ANOVA as the statistical test to compare the HBR between time periods (and/or locations). However, the results of no such test is reported later in the manuscript. See my general comments above.

15.   On page 4, line 152, what do you mean by “Globally”?

16.   In Table 1, in the legend, you state that “Numbers in baskets”. Do you mean “Numbers in parentheses”.

17.   The data in Table 1 could be used in a variety of ways beyond what you have done. A figure could be conducted showing each mosquito species on the x axis, which the numbers collected on the y-axis, using symbols to differentiate between years, different colors between locations. The different sampling effort makes comparison more difficult, so perhaps percentages might be easier to compare. Alternatively, one could use stacked barplots to show how the composition of mosquitoes changes between time points and locations. Alpha and beta diversity and other diversity indices could be calculated. Finally, multivariate methods could be used, such as Non-metric multidimensional scaling (NMDS) to visualize the differences in composition between time-points and locations. MANOVA and ANOSIM could be used to test null hypotheses about no significant differences between years, the periods before and during dam construction, the difference between the two locations, and an interaction between time periods and location (i.e. significant difference between the two time periods for Nyabessan but not for Olama). These could be easily done with the current data by a biostatistician using the free statistical software R, providing even better results.

18.   On page 6, line 158, you list the percentages of the different mosquito species collected before the dam construction. Are these calculated from Table 1 for years 2000-2006? If yes, are they calculated across sampling events, corrected for the different person/night/collection, or are these just the averages of the percentages for each year. I believe the first method would be more appropriate.

19.   On page 6, line 159, you write “scares”. Do you mean “scarce”?

20.   On page 6, line 162, you list a p-value of 10^-4. What test does this p-value relate to? Is this a chi-square test, if so, it is just for An. Gambiae s.l. or is it for all species? If the earlier, why don’t you compare each species statistically between the two periods? A logistic regression, or even a binomial test could do that for each species.

21.   On page 6, lines 168-173, you discuss the species composition for Olama. A. moucheti seems to have a higher proportion overall in Olama both before and during the dam construction. Please point this out. This could also be tested statistically, see methods mentioned above.

22.   On page 6, line 174, you start section “3.2 Night biting rates and biting cycles”. This section seems to only report results for Nyabessa, while section 3.3 is reporting the same results for Olama. Please make this explicit in the titles of these sections.

23.   What is the difference between the Human Biting Rate and the species composition? Do we get HBR by simply dividing the number of mosquitoes collected by the number of people who volunteered to collect mosquitoes? If no, what is the difference? Did all mosquitoes that tried to bite get collected?

24.   Section 3.2 and 3.3 is really missing statistical results, including the ANOVAs promised in the methods.

25.   On page 6, line 189-191, you state that the peaks of nocturnal activity moved from 7-9 pm and 1-3 am before the dam construction and to 9-11 pm and 11-3 am during dam construction. To me, these don’t seem to be that different. Is this really a substantive change?

26.   On page 7, Table 2 could again be turned into a set of figures, with one for each entomological index. Each of these figures would have two subplots, one before and one during the dam construction. The x axis could be the species, the y-axis the entomological index, with different symbols showing different years. It would be much easier to interpret than this table.

27.   On page 11, line 216, please make the title of section 3.3. more descriptive.

28.   On page 12, Table 3, see my comments for Table 2.

29.   On page 14, section 3.4, you provide many statements (e.g. line 248: “EIR increase…:”) where having statistical results would strengthen your arguments considerably, so that no-one could say that what you found was only reflecting the specific houses you surveyed, or was just a random fluke. Please do some tests!

30.   On page 14, line 269, you write “during 3 years during”. You don’t need during twice.

31.   On page 15, you again have many statements that would be much stronger if you had statistical comparisons (e.g. malaria vector biting rates were somewhat higher outdoor versus indoors….suggesting an increase in outdoor Plasmodium transmission”. Wouldn’t it be nice to be able to write: “We’ve found a significantly higher Plasmodium transmission outdoors during dam construction.” You have the numbers to prove that, you just need to run the tests and report them.

32.   On page 15, line 300, write “previous studies” instead of “previously studies”

33.   On page 15, line 305-307, you talk about the ecology of A. moucheti. This section might also go into the introduction. What is the life history of the other species, particularly A. paludis? This would be interesting as that is the species that increased considerably in 2016 and contributed to increased malaria transmission risk. How does that relate to the construction of the dam?

34.   On page 15, line 306, you don’t need to capitalize “Lentic Rivers”.

35.   On page 15, line 316, write in “peri urban areas in sub-Saharan Africa”.

36.   On page 15, line 319, write “found” instead of “founded”.

37.   On page 15, line 321, write “new” instead of “newly”.

38.   The Conclusion section on page 15 have many statements that are currently unsupported but would be if you would do the statistical tests (e.g. composition similar between the two locations, dam construction leading to increase may lead to increase in malaria cases, increase in outdoor biting behavior).

39.   On page 16, the first reference, you don’t need space in “Mac Cormack”.

40.   For references 8 & 9, please remove “<<” and “>>”

41.   For reference 19, please don’t underline the journal.

42.   On page 17, for reference 42, remove the “a” and “b”.

Sincerely,

Author Response

We are grateful for your comments and suggestions especially about statistical analysis. These suggestions helped to better present the data and improve the manuscript.

General comments

Reviewer: One might say that the numbers seek for themselves. However, as detailed on the bottom of page 3, mosquitoes were collected in one house in each collection unit in each village at every surveillance event. Descriptive statistics, such as those presented here, can only summarize the situation in each of these respected houses. However, I would presume that these houses were selected as representative samples of the population of interest, which is all the houses in each collection unit in each village. If you would like to make claims relative to all houses, which it sounds like you do later in the manuscript

Authors: Actually, effort was made to select the houses that would be representative of the mosquito and human populations. This sampling method is common in mosquito surveys since they can fly on a distance up to 5 kilometers.

Reviewer: I believe you could improve the quality of your presentation significantly by turning Table 1, Table 2 and Table 3 into figures, and combining Figure 2-5 into a single Figure, as well as the subplots of Figure 6 into a single plot.

Authors: Statistical analyses have performed and the presentation of the results revised accordingly. Below are the details.

Specific comments

1.       In the Abstract, page 1, line 25, please do not leave a space between “12, 189”. Just have “12,189”. Same goes on line 26 to “6, 062”. These occur again later, so please check and correct throughout the manuscript.

Authors: The space has been removed, lines 25 and 26.

2.       In the Abstract, page 1, line 28, you have “16-50 bites/man/night-b/p/n”. Do you mean “16-50 bites/person/night”? If yes, please use that. This occurs throughout the manuscript, please check and correct.

Authors: The revision has been made

3.       In the Abstract, page 1, line 29-30, it sounds like both A moucheti and A. ovengensis, as well as A. gambiae s.l. and A. paludis experience a reduction in their role in disease transmission. This is not what is shown later. Please correct this sentence to reflect the results later on.

Authors: The sentence has been corrected

4.       On page 2, line 52, please add the year to which these statistics on malaria case numbers refer to.

Authors: The year to which these statistics on malaria refer to has been included and the reference updated

5.       On page 2, line 57, please put a comma after “Meanwhile”.

Authors: The comma has been added

6.       On page 2, line 60 and 61 seems to be missing “over” or “more than” when referring to the threshold in terms of storage capacity and spilling capacity for large dams.

Authors: “More than” has been added

7.       On page 2, line 62, which year does the estimate of the number of people at risk in terms of malaria due to dams refer to?

Authors: The year of the estimates of people at risk of malaria due to dam has been included

8.       On page 2, line 80, what is the correct spelling for the country? “Cameroon” or “Cameroun”?

Authors: The spelling has been corrected

9.       On Figure 1, could you please indicate the precise location of the dam?

Authors: The location of the dam has been added on figure 1

10.   On page 3, on the bottom, you describe the protocol for mosquito collection. This seems to be a form of stratified random sampling. Do you agree? If yes, please state that.

Authors: Yes this is a form of stratified random sampling for to ensure the representability of different micro strata of each study sites.

11.   On page 4, line 127-129, you describe the ELISA method that you used to determine the infection status of mosquitoes with Plasmodium falciparum. Does this measure the current infection status of mosquitoes, or the infection history of these mosquitoes, since it measures antibodies? Or does it not matter because mosquitoes never clear the infection?

Authors: The ELISA CSP determines the infection status of mosquitoes at the time of collection. The method has been calibrated, taking into consideration all possible false positives.

12.   On page 4, line 133, you talk about the statistical analysis, and the entomological indicators that you calculated. The first one you mention as “man biting rate (m.a.)”. However, elsewhere you refer to this as the Human Biting Rate (HBR). I believe this later is more widespread in the literature, so I suggest you to use that consistently. Please check and correct throughout.

Authors: Corrections have been made

13.   On page 4, line 139, you mention that you conducted a chi-square test on the number of mosquitoes collected. However, I wonder if that is the appropriate test, given that the number of some species is very low. A Fisher’s exact test might be more appropriate to test the null hypothesis that the proportion of each species is the same between the two periods (2000-2006) and (2014-2016).

Authors: A 2-sample test for equality of proportions has been applied and section on statistical analysis revised accordingly.

14.   On page 4, line 141, you mention an ANOVA as the statistical test to compare the HBR between time periods (and/or locations). However, the results of no such test is reported later in the manuscript. See my general comments above.

Authors: The results of ANOVA have now been included.

15.   On page 4, line 152, what do you mean by “Globally”?

Authors: Globally has been deleted.

16.   In Table 1, in the legend, you state that “Numbers in baskets”. Do you mean “Numbers in parentheses”.

Authors: “Numbers in baskets” has been replaced by “Numbers in brackets”.

17.   The data in Table 1 could be used in a variety of ways beyond what you have done. A figure could be conducted showing each mosquito species on the x axis, which the numbers collected on the y-axis, using symbols to differentiate between years, different colors between locations. The different sampling effort makes comparison more difficult, so perhaps percentages might be easier to compare. Alternatively, one could use stacked barplots to show how the composition of mosquitoes changes between time points and locations. Alpha and beta diversity and other diversity indices could be calculated. Finally, multivariate methods could be used, such as Non-metric multidimensional scaling (NMDS) to visualize the differences in composition between time-points and locations. MANOVA and ANOSIM could be used to test null hypotheses about no significant differences between years, the periods before and during dam construction, the difference between the two locations, and an interaction between time periods and location (i.e. significant difference between the two time periods for Nyabessan but not for Olama). These could be easily done with the current data by a biostatistician using the free statistical software R, providing even better results.

Authors: Data in Table 1 have been presented in figure 2.

18.   On page 6, line 158, you list the percentages of the different mosquito species collected before the dam construction. Are these calculated from Table 1 for years 2000-2006? If yes, are they calculated across sampling events, corrected for the different person/night/collection, or are these just the averages of the percentages for each year. I believe the first method would be more appropriate.

Authors: They are calculated across sampling events

19.   On page 6, line 159, you write “scares”. Do you mean “scarce”?

Authors: Yes we mean “scarce”, the revision has been made.

20.   On page 6, line 162, you list a p-value of 10^-4. What test does this p-value relate to? Is this a chi-square test, if so, it is just for An. Gambiae s.l. or is it for all species? If the earlier, why don’t you compare each species statistically between the two periods? A logistic regression, or even a binomial test could do that for each species.

Authors: Each species has now been compared between the two periods in Table 1 and the text revised accordingly

21.   On page 6, lines 168-173, you discuss the species composition for Olama. A. moucheti seems to have a higher proportion overall in Olama both before and during the dam construction. Please point this out. This could also be tested statistically, see methods mentioned above.

Authors: The 2-sample test for equality of proportions has also been applied to Olama (Table 1) and the text revised accordingly

22.   On page 6, line 174, you start section “3.2 Night biting rates and biting cycles”. This section seems to only report results for Nyabessa, while section 3.3 is reporting the same results for Olama. Please make this explicit in the titles of these sections.

Authors: The title of the section and the headings have been revised

23.   What is the difference between the Human Biting Rate and the species composition? Do we get HBR by simply dividing the number of mosquitoes collected by the number of people who volunteered to collect mosquitoes? If no, what is the difference? Did all mosquitoes that tried to bite get collected?

Authors: The Human Biting Rate is equivalent to the number of bites that might receive one person per night which is actually, dividing the number of mosquitoes collected by the number of people who volunteered to collect mosquitoes divided by the total number of collection nights. The species composition is the proportions of each mosquito species among the collected samples. The volunteers were trained and evaluated before there enrollment in the study and they were rotating between indoor and outdoor every two hours. This was to increase the chance to collect all the mosquitoes trying to bite them and avoid bias due to individual attractiveness.

24.   Section 3.2 and 3.3 is really missing statistical results, including the ANOVAs promised in the methods.

Authors: See above

25.   On page 6, line 189-191, you state that the peaks of nocturnal activity moved from 7-9 pm and 1-3 am before the dam construction and to 9-11 pm and 11-3 am during dam construction. To me, these don’t seem to be that different. Is this really a substantive change?

Authors: The data have been analyzed and the text revised according to the new figures. In general, the peaks are between 11:00 pm and 01: am or between 01:00 am and 03: am. However in 2016, the peaks of biting activity have moved to 3:00 – 05:00, which might indicate a change of biting cycle at that time.

26.   On page 7, Table 2 could again be turned into a set of figures, with one for each entomological index. Each of these figures would have two subplots, one before and one during the dam construction. The x axis could be the species, the y-axis the entomological index, with different symbols showing different years. It would be much easier to interpret than this table.

Authors: The data from table 2 have been reanalyzed and used to build figures as suggested.

27.   On page 11, line 216, please make the title of section 3.3. more descriptive.

Authors: The title has been revised

28.   On page 12, Table 3, see my comments for Table 2.

Authors: See above

29.   On page 14, section 3.4, you provide many statements (e.g. line 248: “EIR increase…:”) where having statistical results would strengthen your arguments considerably, so that no-one could say that what you found was only reflecting the specific houses you surveyed, or was just a random fluke. Please do some tests!

Authors: Statistical analyses have been performed and the text revised accordingly.

30.   On page 14, line 269, you write “during 3 years during”. You don’t need during twice.

Authors: The text has been changed completely.

31.   On page 15, you again have many statements that would be much stronger if you had statistical comparisons (e.g. malaria vector biting rates were somewhat higher outdoor versus indoors….suggesting an increase in outdoor Plasmodium transmission”. Wouldn’t it be nice to be able to write: “We’ve found a significantly higher Plasmodium transmission outdoors during dam construction?” You have the numbers to prove that, you just need to run the tests and report them.

Authors: The text has been revised accordingly.

32.   On page 15, line 300, write “previous studies” instead of “previously studies”

Authors: The revision has been made.

33.   On page 15, line 305-307, you talk about the ecology of A. moucheti. This section might also go into the introduction. What is the life history of the other species, particularly A. paludis? This would be interesting as that is the species that increased considerably in 2016 and contributed to increased malaria transmission risk. How does that relate to the construction of the dam?

Authors: Yes, this section could also go into the introduction. However, we thought this section would be  more relevant in the discussion to support the findings. A paragraph on A. paludis has been added in the discussion.

34.   On page 15, line 306, you don’t need to capitalize “Lentic Rivers”.

Authors: The words have been revised.

35.   On page 15, line 316, write in “peri urban areas in sub-Saharan Africa”.

Authors: The sentence has been reworded.

36.   On page 15, line 319, write “found” instead of “founded”.

Authors: The word has been revised.

37.   On page 15, line 321, write “new” instead of “newly”.

Authors: The word has been revised.

38.   The Conclusion section on page 15 have many statements that are currently unsupported but would be if you would do the statistical tests (e.g. composition similar between the two locations, dam construction leading to increase may lead to increase in malaria cases, increase in outdoor biting behavior).

Authors: See above

39.   On page 16, the first reference, you don’t need space in “Mac Cormack”.

Authors: The name has been revised.

40.   For references 8 & 9, please remove “<<” and “>>”

Authors: These references have been revised.

41.   For reference 19, please don’t underline the journal.

Authors: This reference has been revised.

42.   On page 17, for reference 42, remove the “a” and “b”.

Authors: These references have been revised.

Thank you for the review.

Reviewer 2 Report

In malaria endemic areas important socio environmental changes are underway. The implementation of new infrastructure can change radically ecological habitats of both pathogens and vectors and therefore influencing diseases transmission dynamic. Thus the study proposed by Mbakop et al. is very important for the future of malaria control as we know that in Sub Sahara Africa, the development of infrastructure network (electrification, dam construction….) is changing the rural environment that could impact anopheles vectors dynamic and malaria transmission. However, there are some issues with the methodology used: Major remarks: The authors are comparing anopheles species composition before and during dam construction. However, they did not compare the vectors dynamic according their seasonality. It is well known that according to the period of the year, the mosquitoes abundance can change whether it is the dry or the rainy season. In the results presented, for both sites of study, and during each year of survey, the author did not take into account the influence of climate on anopheles densities and occurrence. For example, before dam construction, the data were collected essentially during the rainy season while during the period of construction, the collections were made in the dry season in 2014 and 2015. Therefore, it become difficult to attribute the changes observed in vector composition and aggressiveness to dam construction. Also, to Olama was chosen as a control site, but the periods of collection in Olama and Nyabessan are different. It would be more representative to make the sampling at the same time.   In rural areas the typologie of breeding sites can determine the dynamic and anopheles species composition. The rainfall intensity could have a great influence on these parameters. Did the authors try to see if rainfall intensity was constant during all over the study period? Also, Olam was considered as control site, and is located in the equatorial region as Nyabessan, but these to site are about 200km apart, is the rainfall intensity the same between these two areas? On another hand, the collection of anopheles before the dam construction ended at 2006, six year before the beginning of the dam construction. How can the authors be sure that the effect observed is due to the implementation of the dam? The authors in the methodology should take into account the implementation of vector control tools such as LLINs. Although the authors discuss this in the discussion section, it is important to know if nets are used and to what degree of coverage. This is important because the implementation of LLINs can influence and even change the composition of the anopheles species composition and dynamic. In addition, it could influence the outdoor and indoor biting pattern. Minors Remarks Line 101: the collection was done specially in Oveng during all the study period or only during 2000 and 2001. The authors must be clear in the sampling area Line 84-85: remove unpublished data and consider “collected data” Table 1: 2002-2003, the data of these two years should be separated, or the authors must clear this. Is Feb-Jul 2002 or 2003? In fig 2, 3 and 4, the outdoor biting is not represented in some graphes. And the authors should adopt the same pattern of representing the graphs in each figure.

Author Response

We are grateful for your comments and suggestions. These suggestions helped to improve the manuscript.

Major remarks:

Reviewer: The authors are comparing anopheles species composition before and during dam construction. However, they did not compare the vectors dynamic according their seasonality. It is well known that according to the period of the year, the mosquitoes abundance can change whether it is the dry or the rainy season. In the results presented, for both sites of study, and during each year of survey, the author did not take into account the influence of climate on anopheles densities and occurrence. For example, before dam construction, the data were collected essentially during the rainy season while during the period of construction, the collections were made in the dry season in 2014 and 2015. Therefore, it become difficult to attribute the changes observed in vector composition and aggressiveness to dam construction.

Author: Collecting data on vector dynamics would bring much more information on the changes of malaria entomological and epidemiological profile, taking into account the climate and other environmental factors. However, our objective was not to study the dynamics of malaria vectors in the study sites. Therefore, the protocol was not designed for this purpose. Our aim was to capture any malaria change in vector population through cross sectional surveys, and we tried to collect the mosquitoes during high transmission periods of the year. Using this approach, we found that A. paludis is becoming a major vector alongside An. gambiae and other vectors in the study area. The relationship with the construction of the dam comes from the bio ecology of this species and the historical scarceness of this species during longitudinal studies conducted in this area before dam Dam construction. From these findings, we should be able to follow the local vector populations after the impoundment of the dam and suggest appropriate vector control measures.

Reviewer: Also, to Olama was chosen as a control site, but the periods of collection in Olama and Nyabessan are different. It would be more representative to make the sampling at the same time.   In rural areas the typologie of breeding sites can determine the dynamic and anopheles species composition. The rainfall intensity could have a great influence on these parameters. Did the authors try to see if rainfall intensity was constant during all over the study period?

Author: The aim of this reserahc was not to study the bio ecology of the malaria vectors which is already known. We tried to diversify the periods of collection across the study period, which was actually very long and could not allow us to conduct surveys during all the seasons from 2000 to 2016.

Reviewer: Also, Olam was considered as control site, and is located in the equatorial region as Nyabessan, but these to site are about 200km apart, is the rainfall intensity the same between these two areas?

Author: The two study sites share the same climatic facies. We think Olama was a good control because the same vector species were collected in the two study sites. Anopheles moucheti was predominant in both sites before dam construction in Nyabessan.

Reviewer: On another hand, the collection of anopheles before the dam construction ended at 2006, six year before the beginning of the dam construction. How can the authors be sure that the effect observed is due to the implementation of the dam?

Author: It appears that the changes in vector composition and infection appear during dam construct and not before. That is why we think this activity might influence the distribution and the behavior of malaria vector population.

Reviewer: The authors in the methodology should take into account the implementation of vector control tools such as LLINs. Although the authors discuss this in the discussion section, it is important to know if nets are used and to what degree of coverage. This is important because the implementation of LLINs can influence and even change the composition of the anopheles species composition and dynamic. In addition, it could influence the outdoor and indoor biting pattern. Minors Remarks Line 101: the collection was done specially in Oveng during all the study period or only during 2000 and 2001.

Author: Although we did not conduct a survey on the use of nets, it is anticipated that LLINs coverage is high in the study areas, because two nationwide LLIN distribution campaigns were conducted in Cameroon in 2011 and 2016. This has been mentioned in the first paragraph of the discussion. But high coverage of LLINs does not remove the effects of environmental modifications on malaria transmission.

Reviewer: The authors must be clear in the sampling area Line 84-85: remove unpublished data and consider “collected data” Table 1: 2002-2003, the data of these two years should be separated, or the authors must clear this.

Author: The data have been collected in 2002 and 2003 have been separated and used to draw the figures.

Is Feb-Jul 2002 or 2003? In fig 2, 3 and 4, the outdoor biting is not represented in some graphes. And the authors should adopt the same pattern of representing the graphs in each figure.

Author: All these issue have been addressed.

Thank you for the review.

Reviewer 3 Report

The authors mention ANOVA and Chi-square analyses in their methods but I do not see the results of those analyses presented anywhere in the manuscript.  This alone is grounds for rejection.

I believe that the phrase "Memve’ele dam" is a proper noun and as such "dam" should be "Dam"

Line 61: "most dams less than 15 m high"

Line 75: Are you using one letter abbreviations or two letter abbreviations?  Either is fine with me but be consistent.

Lines 86 & 87 - I think there are some capitalization issues.

Line 159:  "scarce" not "scares".

Line 161: "anophelines"

Lines 168 - 169: "composition of the anopheline fauna"

Line 170: Don't start a sentence with an abbreviation.

Lines 179-180: "indoors and outdoors"

Lines 184, 189, 195, 212 & 214: Don't start a sentence with an abbreviation.

Line 228: remove semicolon.

Line 278: You have a double negative here.  Either "not recorded either in" or recorded neither in".  To write "not recorded in neither" actually implies it was recorded in both!

Line 306: "lentic rivers"

Line 328: "its" not "its'"

Author Response

We are grateful for your comments and suggestions. These suggestions helped to improve the manuscript.

Reviewer: The authors mention ANOVA and Chi-square analyses in their methods but I do not see the results of those analyses presented anywhere in the manuscript.  This alone is grounds for rejection.

Author: the statistical analyses have been performed and the results section revised accordingly.

Reviewer:: I believe that the phrase "Memve’ele dam" is a proper noun and as such "dam" should be "Dam"

Author: “Memve’ele” is the proper noun and “dam” is “barrage”; that is why we did not write “Dam”.

Reviewer:Line 61: "most dams less than 15 m high"

Author: this was defined by ICOLD Ref 21.

Reviewer: Line 75: Are you using one letter abbreviations or two letter abbreviations?  Either is fine with me but be consistent.

Author: we chose the one letter abbreviation. The text has been revised accordingly.

Reviewer: Lines 86 & 87 - I think there are some capitalization issues.

Author: a capital letter has been added

Reviewer: Line 159:  "scarce" not "scares".

Author: line 161, “scares” has been replaced by “scarce”

Reviewer: Line 161: "anophelines"

Author: line 196, “s” has been added to “anopheline”

Reviewer: Lines 168 - 169: "composition of the anopheline fauna"

Author: lines 193 “mosquitoes” has been replace by “anopheline fauna”

Reviewer: Line 170: Don't start a sentence with an abbreviation.

Author: line 196, “The species” has been added before the abbreviation.

Reviewer: Lines 179-180: "indoors and outdoors"

Author: line 213, "indoors and outdoors" has been added

Reviewer: Lines 184, 189, 195, 212 & 214: Don't start a sentence with an abbreviation.

Author: the text has been revised accordingly.

Reviewer: Line 228: remove semicolon.

Author: the paragraph has been changed.

Reviewer: Line 278: You have a double negative here.  Either "not recorded either in" or recorded neither in".  To write "not recorded in neither" actually implies it was recorded in both!

Author: lines 389-390, the wording has been corrected

Reviewer: Line 306: "lentic rivers"

Author: line 420, the wording has been corrected

Reviewer: Line 328: "its" not "its'"

Author: line 465, the ´ has been removed.

Thank you for the review.

Reviewer 4 Report

Article entitled- Malaria transmission around the Memve’ele hydroelectric dam in South Cameroon: a combined retrospective and prospective study, 2000-2016 has the following major issues-

1. Title and introduction: The type and size of constructed Dam is not associated with mosquito infectivity and malaria transmission. It is not clear why author emphasizing the size (Reference No. 21 and 22) and hydroelectric dam in the title. Unnecessary and unrelated parameters make the manuscript vast but it proved pointless. Introduction is very vague and lack of clear objective/s of the study is missing.

2. Authors did not mention, how did they identify that whatsoever the bites scars were found on the individuals body was due to which particular Anopheles species (A. gambiae, A. ovengensis, A. paludis, A. marshallii, or A. ziemanni) among these many species.

3. Method number 2.4. Molecular Identification 2.5. Determination of Plasmodium falciparum infectivity seems to me irrelevant with the provided results. Author/s should provide the detailed method viz. primer sequence for species identification for each Anopheles mosquito species, how did they perform the ELISA in detail etc.

4. There is inconsistency and differences in results, table and figures at several places. For example: "Nyabessan, A. moucheti biting activity varied between 2 and 14 bites/man/hour overnight, indoor or outdoor, except in 2003 when it increased to 22 bites/man/hour (Figure 3)" is not matching with 1000-2000 bites/man. This error is consistent with almost every mosquito species in year 2000-2003.

 5. About 25% of cited references are not according to the journals guidelines and it is highly recommended that try to cite the English language journals.

Author Response

We are grateful for your comments and suggestions. These suggestions helped to improve the manuscript.

Point 1: Title and introduction: The type and size of constructed Dam is not associated with mosquito infectivity and malaria transmission. It is not clear why author emphasizing the size (Reference No. 21 and 22) and hydroelectric dam in the title. Unnecessary and unrelated parameters make the manuscript vast but it proved pointless. Introduction is very vague and lack of clear objective/s of the study is missing.

Response 1: Previous studies have shown the impact of dam construction on malaria transmission. In order to better describe ecological context of our study, it appears necessary to provide the relevant information on the size of the dam in accordance with the classification of ICOLD (Reference 21). This information was given in the introduction. The objective of the study was “to assess the evolution of malaria entomological indicators around the Memve’ele dam” as said in the introduction.

Point 2: Authors did not mention, how did they identify that whatsoever the bites scars were found on the individuals body was due to which particular Anopheles species (A. gambiae, A. ovengensis, A. paludis, A. marshallii, or A. ziemanni) among these many species.

Response 2:  Each volunteer mosquito collector collected all the mosquitoes landing on his legs and the research team has to identify the collected samples using standard identification keys.

Point 3:  Method number 2.4. Molecular Identification 2.5. Determination of Plasmodium falciparum infectivity seems to me irrelevant with the provided results. Author/s should provide the detailed method viz. primer sequence for species identification for each Anopheles mosquito species, how did they perform the ELISA in detail etc.

Response 3:  Since the relevant references were provided, we think all the details can easily be obtained from the published papers.

Point 4:  There is inconsistency and differences in results, table and figures at several places. For example: "Nyabessan, A. moucheti biting activity varied between 2 and 14 bites/man/hour overnight, indoor or outdoor, except in 2003 when it increased to 22 bites/man/hour (Figure 3)" is not matching with 1000-2000 bites/man. This error is consistent with almost every mosquito species in year 2000-2003.

Response 4:  the results section has completely been revised following the statistical analyses. We made an effort to improve the text.

Point 5:  About 25% of cited references are not according to the journals guidelines and it is highly recommended that try to cite the English language journals.

Response 5:  The list of references has been reformatted.

Thank you for the review.

Round 2

Reviewer 1 Report

Thank you very much for your resubmission. I can tell that you invested a lot of time into this revision, and took my comments and suggestions seriously, which I very much appreciate. I see that you included an additional co-authors, who must be a statistician, and used the R statistical framework for your analysis. As a consequence, I see that your manuscript has much improved and will provide much needed information on the effects of dam construction on malaria transmission risk in sub-Saharan Africa. However, there are a number of small and some more substantive issues that I would like you to further revise to make sure your manuscript is as sound as possible. The biggest issue is the choice of statistical procedures. In particular, the choice of an ANOVA to test the null hypothesis that the Infection Rate is the same between species, before and during dam construction, and between the locations, is deeply problematic. ANOVA assumes that the response variable (in this case Infection Rate) is normally distributed, which is not true for rates, such as the Infection Rate. This is especially true when rates are close to 0 or 1, such as can be seen on Figure 6. Because of this, one cannot trust the results in Table 3. The proper method in this case would be a logistic regression instead. For the other two measures, this is not so surely a problem, but it would need to be investigated such as by looking at diagnostic plots. Please see my detailed comments below:

1.       I applaud the assistance of Dr. Feshu Nana Betrand, you new co-author. However, you didn’t specify his/her affiliation. Could you please add that?

2.       On page 3, line 109, change the sentence for clarity to “Olama, referred to as the control village,…”

3.       On page 4, line 122, you say that periods of mosquito collection and number of person-nights corresponding for each site are reported in Table 1. However, on Table 1, on page 5, it’s not clear that’s what it’s being shown. Are those the number of mosquitoes being collected for each species within each period?

4.       On page 4, line 142, you mention the “m.a”. Is that the same as the Human Biting Rate (HBR)? If it is please have HBR instead for consistency.

5.       On page 4, line 144, you mention “one-way ANOVA”. That would mean that there is a single explanatory variable. However, in Table 2, 3 and 4 you have more than one explanatory variable. Just say ANOVA (if you keep using that).

6.       On page 4, line 158 and 159, you have two p-values (P=0.99) and (P<0.0001), referring to Table 1. I don’t see these p-values in Table 1. Where do they come from?

7.       On page 4, line 162, you state that 35% of the total number of anophelines caught in Nyabessan in 2016. However in Table 1, it is listed as 21.9% between 2014-16. What null hypothesis test is the p-value belonging to?

8.       In Table 1, on page 5, you list 14 p-values of 2-sample test for equality of proportions. Which test did you use? Was it a z-test, a Chi-square test, or a Fisher’s exact test. If it was a z-test, that might be OK for large sample size, but not for small sample size, such as for the more rare species. In addition, if you run so many tests, you have an increased rate of family-wise Type I error, meaning that your results can be significant by mistake. One solution is to use the Bonferroni correction, which would mean that you would use a significance level of 0.0036 instead of 0.05. You would only lose one significant result from this the difference between the proportion of A. paludis in Olama between the two time periods. Another potential solution might be to treat your data as a multivariate dataset, and test the null hypothesis that the composition of the mosquitoes is the same between the two time periods, using a PERMANOVA. You can do this using the vegan package in R, and the adonis function.

9.        I really like Figure 2 on page 6. However why is the resolution so low? You can export of the graph as a TIFF in high resolution. In addition, it looks clunky that the y label has an underscore between the words as “Percent_Composition”. In addition, this graph is broken down into more periods than Table 1, which makes it hard to match and compare. I suggest to make Figure 1 exactly for the numbers in Table 1. That would also help by getting rid of the empty columns which are distracting. Finally, you might want to consider an NMDS plot to visualize the differences in composition between the periods before/during dam construction and the two locations. You can make them using the vegan package in R and the metaMDS function, which has a great tutorial. That would also go great with the PERMANOVA I suggested above.

10.   On page 7, line 208, you mention that “the composition of anopheline fauna was similar to Nyabessan (P>0.09)(Figure 2).” Where does this p-value come from? How does it relate to Figure 2?

11.   On page 7, line 216, you state that for each 6 major vectors, you compared the night biting rates indoors and outdoors. Please list the names of the 6 major vectors here.

12.   On page 7, lines 218-225, you describe the differencesin HBR for different species in different years, as shown on Figure 3. Are these differences statistically significant?

13.   On page 7, line 226-228, you describe the HBR of A. moucheti during the two periods. You refer to Figure 6 here which is the Infection Rate plot. Which plot did you really mean?

14.   At the same location, you have a p-value of 0.04. Where does this come from?

15.   On page 7, line 226, you state that “59 and 125 bites/person/night, respectively”. Do you mean “59 to 125”?

16.   On page 7, line 229, you refer to Figure 5, on the average HBR of other vectors in Olama. However, Figure 5 is comparing biting times indoors and outdoors. Did you mean Figure 3 here?

17.   On page 7, line 236, change “difference” to “different”.

18.   What is exactly being shown on Figure 3? Is this a dotplot, showing the data? Is it showing averages for each year, or is there really just one datapoint for each year? Why is A. ovengensis and A. nivi combined? In Table 1 and  Figure 1 it is separated, but I see that one is only in Nyabessan and one is only in Olama. Instead of this figure, you could show an effect plot, using the “effects” package in R. That would show the mean HBR for each species in each location for each period, as well as the 95% confidence interval for each, using the model that is presented in Table 2. These would allow you to visualize the differences between different species, periods, and locations, and which of these are statistically significantly different. You could then overlay the actual points as you do currently in Figure 3.

19.   Table 2 on page 9 looks like the result of the summary command in R of an lm object. This is nice, but the estimates you report are the partial regression coefficients, describing the impact of each explanatory variable, relative to the reference, while keeping every other explanatory variable constant. It would be much more useful to see the ANOVA table for this model, which you can get with the Anova() function in the Anova package in R, which would test the null hypothesis that there is no difference in the HBR between different levels of each of the explanatory variables, i.e. that the explanatory variable is not important to explain variation in HBR. In addition, it would be very interesting to see if there was significant interaction between different explanatory variables (e.g. is there a difference in the difference in HBR between period at the two different locations). Please try to include all interactions terms (HBR~Period*Location*Species*Position*Hour) if possible. I’m also slightly concerned about pseudo-replication here, because the different hours are all within the same day with the same people. What is the data really like? Do you have one line for each mosquito? Do you have one line for each hour collected? Do you have one line for each day in your dataset? Finally, I’m not convinced that HBR has a normal distribution. Please take a look at the diagnostic plots, and comment on how well the assumptions of ANOVA were violated.

20.   On page 9, line 264, change “Tableau” to “Table”.

21.   On page 9, line 265-266, state that the differences are relative to 7-9 PM.

22.   On page 9, lines 269 to 285, you describe the differences in biting activity in the different species at the two different locations in the two time periods, based on Figure 4 and Figure 5. However, you do not provide any statistical significance to your comparisons, so we don’t know which of these differences are meaningful and which are just random variations. You could get pairwise differences from your ANOVA (linear model) in Table 2 for each these comparisons either by using the TukeyHSD() function, or if you don’t wish to compare everything with everything, by using the emmeans() functions in the emmeans package. This would provide much better evidence to your assertions about the differences in biting rates.

23.   Page 4 and Page 5 are very descriptive, just showing the actual HBR indoors and outdoors at the different time-points for each vector species. As I suggested for Figure 3, you could instead make an effect plot, which would show the estimated mean HBR for each comparison, as well as the 95% confidence interval. You would not be able to compare each year to each year, but you could add the datapoints at the end, using the ggplot package.

24.   On page 12, line 303, you mention Figure 4 in relation to the Infection Rates. However, Infection Rates are depicted on Figure 6.

25.   On page 12, line 301-302, you state that “For A. gambiae and A. moucheti, the IR significantly decreased from 12.2% to 0.0%...”. Is this based solely on Figure 6, or on the model results depicted on Table 3? Regardless, please state the p-value associated with this decrease.

26.   On page 12, line 303, you provide a p-value (p<0.004). What null hypothesis test does this p-value refer to, using what statistical method?

27.   I’m having a hard time connecting the Infection Rate percentages in the text on page 12 with Figure 6. For example, you state that the “IR of A. gambiae s.l. “ in Olama “remained undetectable until 2016.” However, when I look at Figure 6, it looks like that even in 2014 IR was zero. Similarly, you state on line 311 that “ in 2000 the IR of A. ziemanni was 1.3%”. However, when I look at Figure 6, it looks like that was actually true in 2014. Which one is correct then?

28.   For Table 3, on page 12, I’d like to re-iterate my suggestions from Table 2. Please run a logistic regression instead, and don’t show the output of the “summary” function but the output of the “Anova” function from the “Anova” package. Also, please include interaction terms.

29.   For Figure 6, on page 13, I again would like to recommend an effects plot.

30.   On page 14, line 353, please capitalize the word “figure”.

31.   On page 14, line 355, please use “difference” instead of “different”.

32.   On page 14, line 355-356, you state that there was “no significant difference between the study periods (p=0.5)”. However, in Table 4 the p-value associated with the study period is 0.2398. Why are these p-values different?

33.   On page 14, line 364, use “significantly” instead of “significant

34.   On page 14, line 367, use Figure 7 instead of Figure 6.

35.   On page 14, line 369, I believe the EIR of A. ziemanni was 0.08 ib/p/n in 2014, not in 2000.

36.   On page 14, Table 4, I again suggest to show the results of the “Anova()” function instead of the “summary” function currently shown. Also, please include interactions of these terms. Finally, please discuss the diagnostic plots of this model, and whether or not the assumptions of the ANOVA were violated.

37.   For Figure 7, on page 15, I suggest using an effects plot again.

38.   On page 16, line 398-402, you discuss the human biting rates between indoors and outdoors, and different species. There are currently very sparse statistical evidence for the statements. However, with some additional work on the models, and testing of the pairwise comparisons, you could have all the statistical comparisons you would need to substantiate your statements.

39.   On page 16, line 402, did you mean “showed an exophilic tendency” instead of “and”?

Author Response

 Thank you for your comments, which helped to improve the manuscript.

Reviewer 2 Report

According to the objective of the study,to capture any malaria change in vector population through cross sectional surveys, how can the authors know that the density change observed with the different species is not just a  seasonality effect? We know very well that throughout the year the  density of Anopheles species can change depending on whether the collection is done in the rainy or dry season.  To be sure of capturing the change, it would be more appropriate  to compare the densities obtained during the same periods of the year using a cross sectional approach.To be clearer, it would be nice to know if there is not a seasonal  dynamics, at a scale of one year, because you compare  collections made during different seasons of the year  to conclude to a change in the composition / density between  two different periods.This applies also for the control site.

There is a significant gap, 6 years, between the first and second serie of collections. Even if the construction of a bridge can lead to deep changes, it must also be taken into account that over a period of 6 years, anthropogenic  action (use of mosquito nets and other means of control)  as well as variations in climatic parameters may  influence species composition / density. So, how can you evaluate the impact of these facors on the changes on vectors compsotion/densiy you have observed?

Author Response

 Thank you for your comments, which helped to improve the manuscript.

Response to Reviewer 2 Comments

Point 1: According to the objective of the study, to capture any malaria change in vector population through cross sectional surveys, how can the authors know that the density change observed with the different species is not just a  seasonality effect? We know very well that throughout the year the  density of Anopheles species can change depending on whether the collection is done in the rainy or dry season.  To be sure of capturing the change, it would be more appropriate  to compare the densities obtained during the same periods of the year using a cross sectional approach. To be clearer, it would be nice to know if there is not a seasonal  dynamics, at a scale of one year, because you compare  collections made during different seasons of the year  to conclude to a change in the composition / density between  two different periods.  This applies also for the control site.

Response 1: We agree that the density of Anopheles species usually changes depending on the seasons. In Nyabessan and Olama, the climate is characterized by two rainy seasons which alternate with two dry seasons. Actually, it would be nice to collect the data during all the consecutive seasons, to assess the dynamics of vector populations. We tried to take into consideration the influence of the seasonality on vector dynamics and malaria transmission by collecting the samples during the dry and rainy seasons in both study sites. Considering the duration of the study (2000-2016) and the fact that cross sectional surveys were conducted during dry and rainy seasons before and after dam construction, we pooled the data into these two periods, i.e. period 1 before dam construction and period 2 during dam construction. This allowed us to include the data collected during different seasons and compare the patterns before dam construction versus after dam construction. And some variations were observed, between the two periods. We think these changes may result from ecological modifications of mosquito breeding sites, including the environmental modifications resulting from dam construction. Further studies will allow to update the information on the bio-ecology of malaria vectors and seasonal variations in both study sites. Information on periods of mosquito collections has been added in page 4, line 125-127.

Reviewer 3 Report

This is much improved and inclusion of the statistical analyses really helps the reader to understand what is going on.

Some small housekeeping details:

Line 234: The number of bites per person is more than one so use plural, bites.

Line 235: I would write that the HBR was higher for the three species rather than in them.

Figure 3 & Figure 6 captions: Italicize species names.

Table 1: The authors use parentheses, not brackets.

Line 573: Italicize species name.

Author Response

 Thank you for your comments, which helped to improve the manuscript.

Response to Reviewer 3 comments

This is much improved and inclusion of the statistical analyses really helps the reader to understand what is going on.

Some small housekeeping details:

Point 1: Line 234: The number of bites per person is more than one so use plural, bites.

Response 1: “bite” has been replaced by “bites”, page 8, line 247.

Point 2: Line 235: I would write that the HBR was higher for the three species rather than in them.

Response 2: The text has been revised, page 8, Lines 248-254.

Point 3: Figure 3 & Figure 6 captions: Italicize species names.

Response 3. The names of the species have been italicized

Point 4: Table 1: The authors use parentheses, not brackets.

Response 4: The word “brackets” has been replaced by “parentheses”.

Point 5: Line 573: Italicize species name.

Response 5: The name of the species has been italicize.

Reviewer 4 Report

Dear Author/s, the present form of manuscript is now much better then the earlier version. I would like to appreciate the efforts made my authors to significantly improve the manuscript statistically and scientifically and make it meaningful. Now the revised format of paper is looks much better.

Although, the present version of manuscript looks complete and sufficient by itself but as a part of peer-reviewing procedure, I just want to authenticate the findings with proofs for result section 3.3 and 3.4 which have been derived by follow the method number 2.4 (Molecular Identification by PCR) and 2.5 (determination of Pf infectivity by ELISA). 

Out of three main objectives of the manuscript (1. NBR 2. IR and 3. EIR); the last two most relevant results have been produced by follow the above two procedures 2.4 and 2.5; (for which author did not provide much details; they just refer to the cited literature 34-37).

It would be more satisfactory and reasonable for us, If Author can provide here some details and Gel-images for characterizing the bands size of Anopheles species after DNA isolation and some details how did they do the molecular identification for ~12000 Anopheles mosquito. As well as share the optical density sheet results of ELISA for the identification of CSP of Plasmodium. It is suggested to the author kindly provide the same.

However; These results has been merely asked for as apart of reviewing purpose only, not needed to incorporate  in the manuscript (main text) BUT if authors want they can also add these details in supplementary files (not mandatory); as this is a very huge data collection related study. Moreover; The significance of the ecological and entomological study with such a big data set (collection of mosquitoes for 16 years upto ~12000 different Anopheles) would be more pronounce with the supplementation of these results.

Other Points: Affiliation of author named as "Feshu Nana Betrand" is missing. Minimize the addition of so many references which convey the same message; (some places, I found 6 references in the introduction). One statement can be verify through only by citing 1-2 or maximum 3 references. I would suggest to author/s that use/cite only the more appropriate one and recent. Make the uniformity in the referencing and check the English and grammar one more time.

I would recommend to the Author/s continue such studies in future too which are adding the demographical knowledge of vector mosquitoes species during the course of time.

Author Response

(The authors gave the same response as above.)
